# A Generalizable Deep Voxel-Guided Morphometry Algorithm for Change Detection in Multiple Sclerosis

**Anish Raj**[*1,2]                    ANISH.RAJ@MEDMA.UNI-HEIDELBERG.DE
[1] *Computer Assisted Clinical Medicine, Medical Faculty Mannheim, Heidelberg University, Mannheim, Germany*
[2] *Mannheim Institute for Intelligent Systems in Medicine, Medical Faculty Mannheim, Heidelberg University, Mannheim, Germany*

**Achim Gass**[3,4]
**Philipp Eisele**[3,4]
[3] *Department of Neurology, University Medical Centre Mannheim, Medical Faculty Mannheim, Heidelberg University, Mannheim, Germany*
[4] *Mannheim Center for Translational Neurosciences, Medical Faculty Mannheim, Heidelberg University, Mannheim, Germany*

**Andreas Dabringhaus**[5]
[5] *VGMorph GmbH, Mülheim an der Ruhr, Germany*

**Matthias Kraemer**[5,6]
[6] *NeuroCentrum, Grevenbroich, Germany*

**Frank G. Zöllner**[1,2]

**Editors:** Accepted for publication at MIDL 2024

## Abstract

We present a deep learning-based approach to generate Voxel-Guided Morphometry (VGM) maps from MRI scans, aiming to improve the detection and monitoring of Multiple Sclerosis (MS) progression. Leveraging a 3D U-Net architecture with attention mechanisms and optimized by histogram matching, our model excels in processing three diverse datasets. It demonstrates enhanced accuracy in identifying MS-related changes, outperforming the reference method in mean absolute error by an average of 0.4%. Additionally, visual analysis confirmed our method yields more precise and stable VGM maps across all datasets, compared to the reference. This work underscores the potential of deep learning in MS progression and treatment assessment.

**Keywords:** Voxel-guided morphometry, multiple sclerosis, change maps, convolutional neural networks.

## 1. Introduction

Multiple sclerosis (MS) is a progressive neurological disorder marked by demyelination and axonal damage within the central nervous system (Dal-Bianco et al., 2017). The precise detection and monitoring of MS-induced cerebral changes are pivotal for effective disease management. This work introduces a deep learning model to generate Voxel-Guided Morphometry (VGM) maps from longitudinal MRI data, enhancing the analysis of MS progression. Our model, building on existing deep learning methods (Schnurr et al., 2020), is designed for greater generalizability and reliability in predicting across diverse datasets.

---

[*] Corresponding author

## 2. Materials and Methods

### 2.1. Patient data & ground truth generation

**Data:** In this study, we utilized three MS patient datasets from three distinct sources, adhering to the 2010 criteria (Polman et al., 2011): Dataset A (71 patients, same as (Schnurr et al., 2020)), Dataset B (97 patients), and Dataset C (19 patients from (Carass et al., 2017)). Each dataset includes baseline and 12-month follow-up MRI scans. **VGM generation**:

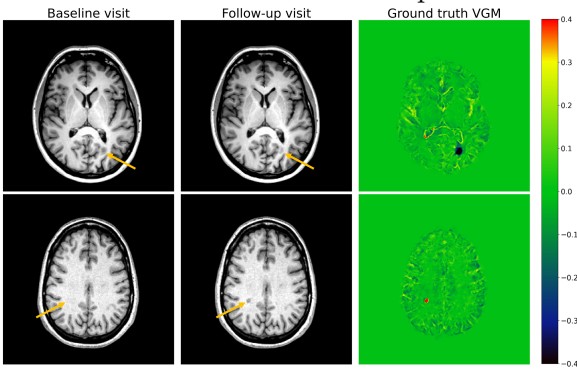

Figure 1: VGM examples: a lesion reduction (top, dark) and a new lesion (bottom, red) with arrows marking lesions.

VGM aligns 3D MRI images and generates maps to detect brain changes between two-time points using T1-w MRI data (Schormann and Kraemer, 2003). VGM guides voxel movement based on grey values to extract volume alterations, resulting in quantified maps that reveal changes in brain regions. Figure 1 depicts ground truth VGM maps.

### 2.2. Image pre-processing

Image normalization was performed by training the Nyul normalizer (Nyúl and Udupa, 1999) with Dataset A and applying it to Datasets A, B, and C. We then resample Dataset B and C to match the voxel spacing of Dataset A (0.94 x 0.94 x 2.00 mm) Additional preprocessing steps can be found in (Raj et al., 2024).

### 2.3. Network architecture & training

We developed two 3D U-Nets, one standard and one enhanced with attention mechanisms to generate VGM maps from longitudinal MRI (i.e. a baseline and follow-up scans). The

Table 1: Dataset metrics: Dataset A from 5-fold training and Dataset B and C from a 5-model ensemble on Dataset A. Bold and underlined numbers signify the best results and significant improvements (p-value < 0.05) over baseline, respectively.

| Dataset | Network | SSIM ↑ | MAE ↓ |
|---|---|---|---|
| Dataset A | U-Net (baseline) | $0.9139 \pm 0.0216$ | $0.0350 \pm 0.0112$ |
| | SE-Attention U-Net | $\mathbf{0.9172 \pm 0.0212}$ | $\mathbf{0.0337 \pm 0.0106}$ |
| Dataset B | U-Net (baseline) | $0.9207 \pm 0.0187$ | $0.0364 \pm 0.0064$ |
| | SE-Attention U-Net | $\underline{\mathbf{0.9416 \pm 0.0143}}$ | $\mathbf{0.0337 \pm 0.0052}$ |
| Dataset C | U-Net (baseline) | $\mathbf{0.9102 \pm 0.0377}$ | $\mathbf{0.0422 \pm 0.0091}$ |
| | SE-Attention U-Net | $0.9091 \pm 0.0291$ | $0.0459 \pm 0.0086$ |

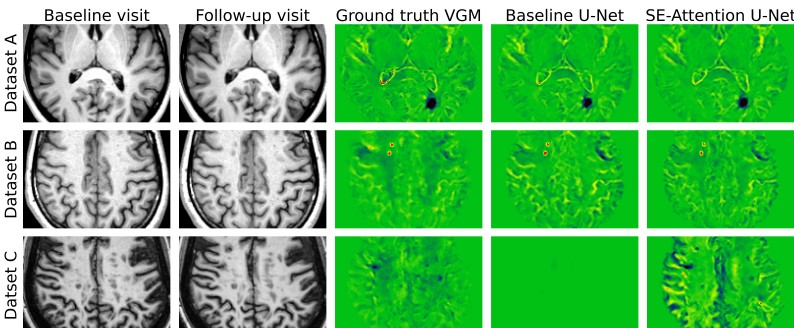

Figure 2: Qualitative comparison between baseline U-Net and our SE-Attention U-Net for one patient per dataset.

first is a baseline U-Net described in (Schnurr et al., 2020)), and the second, is a SE-Attention U-Net equipped with SE and attention blocks in its architecture (Oktay et al., 2018; Hu et al., 2018). The training involved optimizing mean absolute error (MAE) and gradient loss on 3D patches (128x128x32) constrained within brain masks. We employed 5-fold cross-validation on Dataset A, split into train:validation:test subsets (54:2:15). We validated the models' generalizability on Dataset B and C. For the baseline method, we followed the settings described in (Schnurr et al., 2020). Comprehensive architecture and procedural specifics are detailed in our primary publication (Raj et al., 2024).

## 3. Results

Our results depicted in Table 1 show that our SE-Attention U-Net outperforms the baseline U-Net for Datasets A (SSIM of 0.9172, MAE of 0.0337) and B (SSIM of 0.9416, MAE of 0.0337). The improvements are statistically significant (p-value < 0.05). For Dataset C, the baseline outperforms our network. However, visual results show that the baseline values are very close to zero with small variance and so, the VGM map looks empty (cf. Figure 2). Two expert neurologists (A.G., P.E.) visually evaluated the network maps and confirmed that our network produces higher-quality VGM maps. Moreover, inference takes about 2.75 seconds, reducing the total calculation time to 4 minutes (original: 11 minutes). For comprehensive results description, refer to (Raj et al., 2024).

## 4. Discussion

Our goal was to create a fast and generalizable method for predicting VGM maps in MS patients. Across two datasets, our approach consistently outperformed the deep learning reference method (Schnurr et al., 2020) in terms of SSIM and MAE. It was further validated visually and found to be of higher quality across all three datasets. Its independence from MRI sources makes it a versatile clinical tool for detecting subtle MS lesion changes. In summary, our method enables rapid and reliable VGM map generation, enhancing the detection of MS patient brain scan changes.

## Acknowledgments

This research project received funding within the framework "KI für KMU" by the Ministry of Economic Affairs of the State Baden Württemberg, Germany.

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
