# OpenReview forum: "A Generalizable Deep Voxel-Guided Morphometry Algorithm for Change Detection in Multiple Sclerosis"
_MIDL.io/2024/Short_Papers — MIDL 2024 Short Papers_

### Official Review · Reviewer_QU6Q · 2024-04-24

**Confidence:** 4
**Final Rating:** 4

**Review:**

The authors propose to use an attention U-net rather than a vanilla U-Net to generate VGM maps from longitudinal MRI data. Using three datasets they show small but significant improvements on 2 datasets and non-inferior results on another dataset.

Strengths
- The task is clinically relevant
- Performance metrics are improved on 2 datasets
- Method is faster

Weaknesses
- Hyperparameter optimisation is not described, perhaps the baseline could be more competitive
- Neurologist expert opinion on VGM quality should be quantified (e.g. quality scores on a Likert scale) and tested statistically
- Novelty is limited

---

### Decision · Program_Chairs · 2024-04-26

Accept